# ERK is a Pivotal Player of Chemo-Immune-Resistance in Cancer

**DOI:** 10.3390/ijms20102505

**Published:** 2019-05-21

**Authors:** Iris C. Salaroglio, Eleonora Mungo, Elena Gazzano, Joanna Kopecka, Chiara Riganti

**Affiliations:** Department of Oncology, University of Torino, via Santena 5/bis, 10126 Torino, Italy; irischiara.salaroglio@unito.it (I.C.S.); eleonora.mungo@unito.it (E.M.); elena.gazzano@unito.it (E.G.); joanna.kopecka@unito.it (J.K.)

**Keywords:** ERK, chemoresistance, immune-resistance, immune-escape

## Abstract

The extracellular signal-related kinases (ERKs) act as pleiotropic molecules in tumors, where they activate pro-survival pathways leading to cell proliferation and migration, as well as modulate apoptosis, differentiation, and senescence. Given its central role as sensor of extracellular signals, ERK transduction system is widely exploited by cancer cells subjected to environmental stresses, such as chemotherapy and anti-tumor activity of the host immune system. Aggressive tumors have a tremendous ability to adapt and survive in stressing and unfavorable conditions. The simultaneous resistance to chemotherapy and immune system responses is common, and ERK signaling plays a key role in both types of resistance. In this review, we dissect the main ERK-dependent mechanisms and feedback circuitries that simultaneously determine chemoresistance and immune-resistance/immune-escape in cancer cells. We discuss the pros and cons of targeting ERK signaling to induce chemo-immune-sensitization in refractory tumors.

## 1. Introduction

Extracellular signal-regulated kinases (ERKs) 1-5, c-Jun amino-terminal kinases (JNKs) 1-3, and p38 isoforms are part of the so-called mitogen-activated protein kinases (MAPKs) [1], an evolutionary conserved family of proteins controlling mitosis, survival, apoptosis, differentiation, and metabolism [2]. ERK members are mainly activated in response to mitogenic stimuli upon the engagement of tyrosine kinase receptors (TKRs) that transduce their signals through the rat sarcoma 2 viral oncogene homolog/rapidly accelerated fibrosarcoma/MAPK/ERK kinase (RAS/RAF/MEK) axis. The main substrates of ERKs include: plasma-membrane-associated proteins (cluster of differentiation 120, CD120a; spleen tyrosine kinase, Syk; calnexin); transcription factors inducing “early-response genes” (steroid receptor coactivator-1, SRC-1; paired box protein 6, Pax6; nuclear factor of activated T-cells, NFAT); ETS domain-containing protein, Elk-1; myocyte enhancer factor-2, MEF2; activator protein-1 (AP1/c-jun and c-fos); avian myelocytomatosis virus oncogene cellular homolog, c-myc; signal transducer and activator of transcription-3, STAT3; cytoskeletal proteins (paxillin); protein kinases and phosphatase (ribosomal S6 kinase, RSKs; mitogen-and stress-activated kinases, MSKs); and epigenetic modifiers (cytosine-5-methyltransferase 3β, DNMT3B) [1,3,4,5]. Most of the target genes induce the up-regulation of cyclins and cyclin-dependent kinases (CDKs) and the down-regulation of the cell cycle checkpoints regulators p21 and p27 [1,2,3,4,5]. As a result, the MEK/ERK axis increases the proliferation of differentiated cells [1,2,3,4,5], germline stem cells [6], and cancer stem cells [7]. The broad spectrum of substrates makes ERKs key regulators of proliferation, migration, and apoptosis, as well as appealing therapeutic targets in cancer.

Since ERKs are activated in response to extracellular stimuli, ERKs work as unique hubs that finely sense changes in the tumor microenvironment, producing adaptive responses of cancer cell [8,9,10]. RSKs and MSKs are downstream effectors of ERKs strongly involved in such stress adaptation. RSKs modulate cell proliferation, survival, and migration by releasing the active eukaryotic translation initiation factor 2β (eIF2B) from glycogen synthase kinase 3-β (GSK3β), sequestrating complex and increasing protein translation [1], and by activating pro-survival programs under the control of cAMP response element-binding protein (CREB), serum response factor (SRF), nuclear factor-kB (NF-kB), and NFAT3 [11]. MSK1-2 also cooperates to activate the transcriptional programs driven by CREB, AP1, NF-kB, and STAT3 [1]. Moreover, by phosphorylating the histone H3 and the chromatin-associated protein high mobility group-14 (HMG-14), MSK1-2 epigenetically induces the transcription of pro-survival genes [12].

Stressing conditions such as nutrient deprivation, hypoxia, UV, chemotherapy, or radiotherapy [13] are often experienced by rapidly growing tumors. The ability of cancer cells to adapt to unfavorable environmental conditions confers a selective advantage in terms of survival and proliferation over normal cells [14,15]. ERK1/2 has been reported to mediate resistance to chemotherapy in endometrial [16], gastric [17], colon [18], breast [19], ovarian [20], liver [21], oesophageal [22], prostate [23], non small [24] and small cell lung [25] cancers, osteosarcomas [26], neuroblastomas [27], gliomas [28], and T-cell acute lymphoblastic leukemias [29]. ERK1/2 also promotes an immune-evasive phenotype in colon [30], breast [31], prostate [32], liver [33], non small cell lung [34] cancers, pleural malignant mesotheliomas [35], gastrointestinal sarcomas [36], Lewis lung carcinomas and melanomas [37], and glioblastomas [38]. This review focuses on the role of ERKs in mediating the adaptation of cancer cells to chemotherapy and anti-tumor activity of the host immune system, analyzes the ERK-dependent molecular circuitries that confer chemoresistance and immune-resistance/immune-escape, and discusses the pros and cons of ERKs inhibitors as chemo-immune-sensitizers agents.

## 2. ERKs Favor an Aggressive Phenotype in Tumors in Response to Pleiotropic Intrinsic and Extrinsic Stimuli

Specific knocking-down of ERK1 or ERK2 points out that these two kinases have often redundant and overlapping biological functions [2], leading to the hypothesis that the ERK’s global activation—more than the activation of specific isoforms—determines the cell fate.

The mechanisms of ERK1/2 activity depend on the mutational landscape of each tumor.

Oncogenic mutations in upstream activators are common in tumors and determine the constitutive activation of ERKs. For instance, RAS constitutively activating mutations occur in 90% of pancreas cancers, 50% of colon and thyroid cancers, 30% of lung cancers, and 25% of melanomas [39]. Overall, oncogenic missense mutations of RAS concentrated in specific hot spots around glycine 12 and glutamine 16 have been detected in 30% of all cancers [39]. The downstream effector B-RAF is mutated in 7% of all cancers, ranging from 100% of hairy cell leukemias to 50–60% of melanomas, 40–60% of thyroid cancers, and 5–10% of colorectal cancers [40]. The V600E missense mutation is by far the most common mutation activating B-RAF [40,41]. MEK mutations have been detected in melanomas [42], ovary cancers [43], and gliomas [44]. The presence of oncogenic mutations in the components of the RAS/RAF/MEK axis determines an abnormally stable and prolonged activation of ERK instead of its physiological pulsatile activity, inducing the constitutive transcription of pro-proliferative factors such as c-jun, EGR1, and cyclin D1 [45].

According to the Tissue Cancer Genome Atlas (TCGA) available datasets, ERK1 is amplified in 16% of ovarian cancers, 9% of bladder cancers, and 6% of lung and breast cancers, with lower percentages in the other cancer types. The percentage of mutations is always below 1%, with prevalent synonymous or missense mutations that have moderate or low impact inducing no change or low reduction in the kinase activity (available online: https://portal.gdc.cancer.gov/genes/ENSG00000102882). The amplifications of ERK2 are present mainly in ovarian, bladder, lung, and breast cancers with percentages of 21%, 13%, 12%, and 5%, respectively. No more than 2% of all cancer types have missense mutations or deletions in the 3′-untranslated region (UTR) of ERK2 with moderate impact or probably deleterious meaning on the protein activity (available online: https://portal.gdc.cancer.gov/genes/ENSG00000100030). Such a low percentage of deleterious mutations is not surprising. Since ERK is a pro-survival factor, it is expected that cancer cells with deleterious mutations are eliminated, while the clones most adapted to survive are maintained. Surprisingly, few activating mutations have been detected in ERK1 and ERK2. We may speculate that, since many activating mutations are contained in the upstream molecules (such as RAS, BRAF, and MEK), ERKs activating mutations or amplifications are not so crucial to sustain cell proliferation and survival. More than activating mutations or amplifications in the ERK1/2 gene, the abnormal activation of these kinases as a consequence of oncogenic mutations in the RAS/RAF/MEK axis promotes cell survival and adaptation to unfavorable conditions, including chemotherapy and immune system anti-tumor response.

Besides the oncogenic activation in the RAS/RAF/MEK axis, many other factors increase ERK activity in cancer cells.

Exogenous or autocrine growth factors activating RTKs—such as epidermal growth factor receptor (EGFR), receptor tyrosine-protein kinase erbB-2 (Her2), platelet-derived growth factor receptor (PDGFR), insulin-like growth factor 1 (IGF1-R), vascular endothelial growth factor (VEGFR), fms-like tyrosine kinase 3 (Flt3), and c-Kit—increase the activity of downstream ERK1/2 in many cancers [7]. In human neuroblastoma, the leucine-rich repeat-containing G-protein coupled receptor 5 (Lgr5) belonging to the R-spondin receptor family, activates ERK1/2, which, together with the wingless/integrated (Wnt)/β-catenin pathway, contributes to tumor growth [46].

The cell–extracellular matrix (ECM) interactions also activate RAS and ERK1/2 [47]. Integrins play a key role in transducing the ECM-dependent signal. When integrins bind to their respective ligands, focal adhesion kinase (FAK) is recruited and autophosphorylated. This step is necessary to phosphorylate and activate ERK1/2 that promotes proliferation and metastasis [48,49]. Alternatively, the FAK/Src complex can activate the RAS/RAF/MEK axis by recruiting the enhancer Ras proximate 1 (RAP1) and the adaptor growth factor receptor-bound protein 2 (GRB2) [47]. Fibronectin and integrin-β1 have been found to be associated with EGFR in pancreatic cancer; this complex, together with calreticulin, triggers an epithelial-mesenchymal transition (EMT) program by activating the ERK/MAPK pathway, thus favoring the EGF-induced migration and invasion [50].

Additionally, cytokines cooperate in promoting cell migration mediated by ERKs. For instance, myeloid differentiation factor 88 (MyD88) and interleukin-1 receptor–associated kinase (IRAK), two downstream effectors of interleukin-1 (IL-1), co-localize with actin capping protein (ACP) and phosphorylate ERK1/2 at the ruffled border of migrating cells. The MyD88/IRAK/ACP complex is necessary to mediate the IL-1/ERK1/2-mediated migration [51]. IL-33 promotes cellular proliferation and invasion in ovarian cancer by activating ERK1/2, as demonstrated by the complete abrogation of these responses in cells treated with the ERK inhibitor U0126 [52]. Moreover, ERKs mediate proliferation and/or migration in response to IL-8 in pancreatic [53] and cervical [54] cancer, to IL-34 in colon cancer cells [55], and to IL-13 in glioblastoma multiforme [56], working as downstream collectors of the pro-tumorigenic inputs of different cytokines.

By cooperating with the prosurvival pathway phosphoinositide 3-kinase (PI3K)/protein kinase B (AKT), ERK1/2 induces EMT [57] and promotes metastatization by recruiting metalloproteinase 9 in gastric cancer [58], increases migration in non small cell lung cancer [59], rhabomiosarcoma [5], ovarian cancer [60], cervical cancer [61], and hepatocellular carcinoma [62], and prevents apoptosis in renal cancer by up-regulating B-cell lymphoma 2 (Bcl-2) and down-regulating p53, Bcl-2-like protein 4 (Bax), and poly (ADP-ribose) polymerase (PARP) [63]. The same anti-apoptotic effect has been reported in breast cancer, where ERK1/2 cooperates with 5′ AMP-activated protein kinase/mammalian target of rapamycin/p70 S6 kinase (AMPK/mTOR/p70S6K) [64], and in colorectal cancer, where it cooperates with small mothers against decapentaplegic-4 (Smad4) [65].

Among the endogenous factors controlling ERK activity, reactive oxygen species (ROS), calcium, and cytoskeleton-associated proteins are well-documented activators of ERKs. For instance, in breast cancer, ROS activates the PI3K/AKT/ERK axis, up-regulating cyclin D and CDK4,and stimulating the entry into S-phase [66]. Oscillations in intracellular calcium, controlled by K-RAS, increase the ERK-driven tumorigenesis in colorectal cancer [67]. The kinesin family member 15 (KIF15), which controls microtubule assembly and mitosis, activates the MEK/ERK axis, promoting pancreatic adenocarcinoma cell growth [68]. These few examples demonstrate that different and unrelated stimuli converge on ERK. Whatever the stimulus is, the final result is an increased rate of cell proliferation and/or invasion.

In addition to activating factors, endogenous antagonists of RAS/RAF/ERK are known. Among these, GTPase activating proteins (GAPs) specific for RAS mainly act as onco-suppressors by reducing the activation of the RAF/ERK axis [69]. Neurofibromin-1 (NF1) is a RAS-GAP protein whose loss of function induces tumorigenesis in the absence of other alterations in RAS/RAF/ERK pathway [70], as it happens in neurofibromatosis type I [71]. RAS P21 protein activator (RASA1) is another RAS GAP whose deleterious mutations increase the activity of the RAS/RAF/ERK axis and promote cell proliferation in non small cell lung cancers without other oncogenic mutations [72]. Of note, when NF1 and RASA1 are co-mutated, they identify a subtype of non small cell lung cancers with higher sensitivity to MEK inhibitors [72]; this genotype likely identifies a subtype of tumors that do not present other oncogenic mutations in the RAS/RAF/ERK axis and are therein sensitive to the pharmacological inhibitors of this pathway. Dual specificity MAPK phosphatases (DUSP) inhibit the RAS/RAF/ERK axis downstream RAS, since they dephosphorylate a broad spectrum of MAPKs. DUSP5, DUSP6, and DUSP7 are specific inhibitors of ERK1/2 [73]. DUSP5 acts as an oncosuppressor protein by preventing ERK1/2 induced cell proliferation [74]. As far as DUSP6 is concerned, while its ectopic expression reduces cell proliferation in pancreatic [75] and lung [76] cancers, DUSP6 has a pro-oncogenic function in B-cell acute lymphoblastic leukemia [77]. When RAS is oncogenically activated in pre-B cells, it promotes the death of a vast majority of cells; however, the small percentage of surviving cells show a strong up-regulation of DUSP6, induced as a compensatory negative feedback by the increased activity of RAS/ERK1/2 [77]. If this up-regulation promotes growth arrest in solid tumors, in B cell progenitors the effect is the progression towards malignant transformation [77]. This complex scenario highlights an ambiguous role of DUSPs that can act as oncosuppressors or oncogenic molecules, depending on the tumor type, the timing of ERK activation, the presence of additional oncogenic mutations in the RAS/RAF/ERK axis [73].

Another inhibitor of ERK1/2 is the oncosuppressor Erbin, which interacts with the kinase suppressor of Ras-1 (KSR1) and down-regulates the RAS/RAF/MEK/ERK1/2 pathway in colorectal cancer [78]. However, its role in other tumors in unknown.

Several synthetic inhibitors of ERK1/2, such as PD98059, PD184161, VTX-11e, FRI-20, ON-01060 [79], pirfenidone [80], bisphenol A, and 2,2-bis(*p*-hydroxyphenyl)-1,1,1-trichloroethane [81], as well as natural products such as krukovine [82], globularifolin [8], tanshinone IIA [10], reduce cancer cell proliferation in vitro. In rhabdomyosarcoma [5] and oesophageal squamous cell carcinoma [83], the reduced proliferation is paralleled by an increased differentiation, suggesting that ERK inhibition has pleiotropic effects. Most studies on ERK activity, however, do not consider the different oncogenic mutations constitutively activating the RAS/RAF/MEK/ERK1/2 axis and the different timing of decay signals. These are critical factors in determining the efficacy of inhibition. Classical RAF inhibitors such as vemurafenib and SB590885, or the MEK inhibitor U0126, reduce the entry into S-phase if applied to cancer cells with wild-type RAS and RAF that have pulsatile activation of MEK/ERK1/2. By contrast, in cells with a constitutively oncogenic activated RAS/RAF/MEK/ERK1/2 axis, they paradoxically increase the timing of activated ERK1/2 and the proliferation [45]. The absence or the presence of oncogenic mutations in the RAS/RAF/MEK/ERK1/2 axis may account for the different results obtained by ERK1/2 inhibitors. The different pharmacodynamic properties of ERK inhibitors and the intra- and the inter-tumor variability are additional factors that may explain the higly variable efficacy of ERK inhbitors in different studies.

In contrast with the evidence reporting an anti-tumor activity of ERK1/2 inhibitors, it is noteworthy that they may have deleterious consequences in specific cell types, such as peripheral blood monocytes [84], skin squamous cancer cells [85], and colorectal cancer cells [86], where ERKs physiologically induce senescence by impairing telomerase activity. Senescent cells activate the protective autophagic pathway in response to MEK/ERK inhibitors [87]. In this case, targeting ERK may increase the emergence of dormant tumor clones, which are highly drug resistant and hard to eradicate.

## 3. ERKs Favor the Adaptation to Environmental Stresses

ERK1/2 activity promotes the adaptation to oxidative stress [88] and endoplasmic reticulum (ER) stress [14,15] in cancer cells, two conditions that are often induced by environmental stresses, such as chemotherapy and anti-tumor activity of the host immune system. The effects and the mechanisms are highly tumor-dependent.

In neuronal cells, pro-oxidant stimuli promote the activation of the deglycase protein DJ-1 that activates the RAF/MEK/ERK1/2 axis and prevents the activation of the inhibitor DUSP1. ERK1/2 is free to translocate into the nucleus and cooperate with Elk1 in inducing the transcription of the anti-oxidant enzymes superoxide dismutase 1 (SOD1) [89]. In tumor cells, the activation of the RAF/MEK/ERK1/2 axis by DJ-1 and the protection from oxidative stress promote transformation and drug resistance [90].

ERK1/2 activation prevents the damage induced by increased intracellular calcium [91] that elicits the so-called ER stress [14,15]. Depending on the physiopathological situations, this effect could be positive or negative. In cardiac ischemia-reperfusion or in neurodegenerative diseases, when the increase in intracellular calcium damages the cell, the activation of ERK1/2 is protective. In tumors, the release of calcium from ER storages trigger cell apoptosis [14,15]. In this case, ERK1/2 activation protects cancer cells from this ER-dependent damage. The activation of ERK1/2 in response to calcium oscillations is multifaceted. Although ERK1/2 activity counteracts the damages induced by an acute increase in intracellular calcium [45], in colorectal cancer, the oncogenic activation of BRAF/MEK/ERK1/2 induces a chronic ER stress that promotes cell survival and is associated with tumor progression and poor patient outcome [92]. Under nutrients depletion, another condition inducing ER stress, ERK1/2 is activated by the autocrinely produced IL-4 in prostate, breast, ovarian, head, and neck cancers, and contributes to cell survival by activating the anti-apoptotic proteins survivins [93].

If, in most cases, ERK activity helps cells to survive during environmental stress, there is also evidence of the opposite. For instance, the acute exposure to cold down-regulates ERK1/2 and simultaneously activates catabolic pathways driven by AMPK, which inhibits protein synthesis and promotes salvage autophagic mechanisms [94]. Radiotherapy induces cellular damage by producing ROS that induce DNA damage and lipid peroxidation. The increase in lipid peroxidation is favored by the up-regulation of cyclooxygenase 2 (COX2) in irradiated cells, an event that is promoted by ERK1/2 activation [95]. In this case, ERK1/2 activation amplifies the cell damage induced by radiation, while the ERK inhibitor PD98059 has a radio-protective role.

Despite some contrasting evidence, most experimental findings suggest that ERK1/2 protects cancer cells from unfavorable conditions, such as oxidative stress and ER stress. Since these two types of stresses are often induced by chemotherapy, immune-therapy, or the anti-tumor immune response, we analyze and discuss how ERK is implicated in the resistance to chemotherapy- and immune-therapy-based treatments and to anti-tumor activity of the immune system. From this perspective, ERK inhibitors may be indicated as chemo- or immune-sensitizer agents against resistant and aggressive tumors.

## 4. ERKs Mediate the Resistance to Chemotherapy

ERKs mediate chemoresistance with different mechanisms, including reduced apoptosis, increased cell proliferation, and up-regulation of drug efflux transporters (Figure 1).

### 4.1. ERKs Enhance Survival and Prevent Apoptosis in Response to Chemotherapy

In endometrial [16], gastric [17], colon [18], and breast cancers [19], MEK/ERK1/2 causes a multi-drug resistant phenotype by up-regulating anti-apoptotic proteins of the Bcl-2 family. In breast tumors, the mechanism can be reversed by miR-27b, which down-regulates Casitas B-lineage lymphoma proto-oncogene-b (*CBLB*) and *GRB2* genes, two up-stream controllers of ERK [19]. An anti-apoptotic mechanism also mediates the resistance to paclitaxel in estrogen-sensitive breast cancer cells, where ERK1/2 promotes the transcriptional up-regulation of survivins by recruiting p53 on the *survivins* promoter [96].

In osteosarcoma, phosphorylated ERK1/2 induces resistance to cisplatin by up-regulating cyclin D1/E1 and accelerating the entrance into the cell cycle [26]. Similarly, in breast cancer cells, the Aurora A/Src/ERK1/2 axis determines resistance to taxol by increasing the number of cells entering S- and G2-phases, and reducing the number of apoptotic cells [97]. The simultaneous increase in cell proliferation and decrease in apoptosis elicited by ERK1/2 also induces resistance to different chemotherapeutic drugs in ovarian cancer, where ERK1/2 activates the pro-survival effectors mitogen activated kinase kinase (MKK) and eukaryotic translation initiation factor 4E (eIF4E) [20], in hepatocellular carcinoma, where ERK 1/2 is part of the bone-morphogenetic protein 4 (BMP4)-dependent signalling [98], and in colon cancer, where ERK1/2 cooperates with the pro-survival transcription factor NF-kB [99]. ERK and NF-kB synergize in promoting the resistance to anthracyclines in breast cancer, where the transmembrane tumor necrosis factor-α (tmTNF-α) activates both detoxification pathways dependent on ERK/glutathione-S transferase π and anti-apoptotic pathways dependent on ERK/NF-kB [100].

The cooperation of ERKs with other pro-survival pathways and/or with inactivated oncosuppressor factors is very common in chemoresistant tumors. For instance, in oesophageal cancer, the deletion of the pro-apoptotic oncosuppressor receptor interacting protein kinase 3 (RIP3) activates the cell division cycle 37 homolog/heat shock protein 90 (CDC37/HSP90) complex, which in turns activates ERK, JNK, and AKT [21]. All these kinases mediate resistance to cisplatin, as demonstrated by the chemosensitizing effects of their specific pharmacological inhibitors [21]. In chemosensitive prostate tumors, AKT promotes the phosphorylation of the O-class forkhead factor FOXO1, which binds Ras GTPase-activating-like protein IQGAP, a scaffold protein activating multiple MAPKs. This situation prevents the activation of the RAF/MEK/ERK axis mediated by IQGAP. By contrast, in chemoresistant tumors, paclitaxel or PI3K inhibitors induce the nuclear translocation of FOXO1, removing the FOXO1-induced inhibition on IQGAP. These events activate ERK1/2 that induces resistance to paclitaxel [23].

From a translational perspective, these multiple cross-talks open the possibility of using different targeted therapies (e.g., NF-kB inhibitors, FOXO1 phosphomimetics, and BRAF/ERKs inhibitors) as potential chemosensitizer agents. A deep knowledge of the oncogenic alterations present in each tumor is required to choose the proper agent and to move forward with a more personalized treatment.

Decreased apoptosis and increased cell cycle are not the only mechanisms involved in ERK-dependent chemoresistance.

The metabolic profile of cancer cells also plays a role. In breast cancer, the resistance to doxorubicin is associated with the overexpression of fibroblast growth factor receptor 4 (FGFR4), which increases the anaerobic glucose metabolism and activates ERK1/2; both processes determine resistance to doxorubicin, as demonstrated by the chemosensitization elicited by 2-deoxyglucose and the MEK/ERK inhibitor U0126 [101]. Similarly, the overexpression of the glycolityc pace-maker enzyme hexokinase 2 (HK2) mediates the resistance to cisplatin in ovarian cancer by favoring the activation of ERK [102]. In colon cancer [103] and T-cell acute lymphoblastic leukemia [29], ERK1/2 mediates the phosphorylation of dynamin-related protein 1 (Drp1), a factor favoring mitochondrial fission and lowering mitochondrial ROS. This mitochondrial-dependent mechanism protects cells from the oxidative damages induced by chemotherapy. Although these observations do not provide in-depth mechanistic explanations, they are important because most solid tumors use glycolysis as a key energetic pathway and rely on active mitochodria as sources of additional energy and building blocks. Therein, a metabolic reprogramming that uses glycolitytic inhibitors, such as the glucose uptake inhibitor fasentin or the HK inhibitor lonidamine, and mitochondrial oxidative phosphorylation inhibitors, such as metformin, in combination with ERK inhibitors may represent a potential strategy to restore chemosensitivity.

ERKs also mediate chemoresistance in stem cells that are the most chemorefractory component of tumors. For instance, the ERK1/2/p70S6K axis induces resistance to gemcitabine by promoting the proliferation of CD133^+^ pancreatic cancer stem cells. Metformin shows a particular efficacy against CD133^+^-cells, where it counteracts ERK-dependent proliferation and gemcitabine resistance [104]. In non small cell lung cancer stem cells, ERK1/2 activates the GSK3β/β-catenin signaling, increasing both proliferation rate and cisplatin resistance [24]. In ovarian cancer stem cells, ERK1/2 is under the control of amphiregulin/EGFR and mediates both the maintenance of stemness and the resistance to docetaxel and carboplatin [105]. The RNA polymerase II elongation factor (Ell3) induces, at the same time, the expansion of stem-cell-like breast cancer cells and the resistance to 5-fluorouracile in a MEK/ERK1/2-dependent manner [106]. In small cell lung cancer, etoposide expands a population of cells enriched in α2δ1 protein, a Ca^++^ channel that mediates chemoresistance through the activation of ERK1/2 [25], providing an additional mechanism of resistance and of potential therapeutic intervention.

### 4.2. ERKs Mediate Chemoresistance Cooperating with Tumor Microenvironment Factors

The role of the tumor microenvironment is also crucial in determining ERK-mediated chemoresistance.

The interaction with ECM is important in determining resistance to targeted therapies and chemotherapeutic agents. One mechanism by which ECM proteins activate ERK1/2 and mediate chemoresistance is the up-regulation of the Bcl-2 proteins family. This mechanism is active in oesophageal cancer, where the silencing of type I collagen and fibronectin reverses ERK1/2-mediated chemoresistance [107], and in ovarian cancer, where the ECM protein nidogen 1 (NID1) activates ERK1/2, triggering both EMT and cisplatin-resistance [108]. The interaction between laminin-5 and integrin-β1 promotes the resistance to trastuzmab in metastatic breast cancers by activating pro-survival ERK1/2-dependent pathways [109]. Collagen I prevents Fas-mediated apoptosis in the leukemic T-Jurkat cells by stimulating integrin-*α*2*β*1/EK1/2-dependent pathways that reduce the activation of caspase 8 [110]. Similarly, in T-cell acute lymphoblastic leukemia, the interaction of collagen I with integrin-*α*2*β*1 triggers the activation of MAPK/ERK, which determines resistance to doxorubicin-induced apoptosis [111]. Connective tissue growth factor (CTGF), a secreted protein that binds integrins, induces resistance to the apoptosis elicited by cisplatin in osteosarcoma cells by activating the integrin-downstream axis FAK/ERK1/2 pathways [112]. Integrins may also cooperate with other surface receptors of cytokines and chemokines to induce chemoresistance. For instance, in B-cell acute lymphoblastic leukemia, the presence of bone marrow stromal cells up-regulates a plasma-membrane-associated complex constituted by integrin-*α*4*β*1, C-X-C chemokine receptor 4 (CXCR4), and the human ether-a-go-go related gene 1 (hERG1) channel. This trimeric complex transduces through MAPK/ERK and PI3K/AKT pathways, inducing resistance to doxorubicin, methotrexate, and prednisone [113].

Additionally, the tumor-associated stromal cells contribute to increased chemoresistance via stroma-tumor cell contact or growth factors, cytokines, and chemokines secreted in the tumor microenvironment. In neuroblastoma, mesenchymal stromal cells activate the MEK/ERK1/2 axis that induces—together with the Janus kinase 2 (JAK2)/STAT3 pathway—resistance to etoposide [27]. Similarly, bone marrow mesenchymal stromal cells are necessary to activate the ERK/Drp1 axis that determines chemoresistance to T-cell acute lymphoblastic leukemia [29]. Ovarian cancer-associated fibroblasts up-regulate the lipoma-preferred partner (*LPP*) gene in the endothelial cells with a FAK/ERK1/2/CREB-dependent mechanism. LPP reduces the leakiness of tumor-infiltrating capillaries, lowering the intra-tumor delivery of paclitaxel and increasing chemoresistance [114].

IL-34 mediates resistance to oxaliplatin in colon cancer cells in an ERK1/2-dependent manner, as suggested by the cell chemosensitization induced by the pharmacological inhibitors of ERKs [54]. In human glioma, the insulin-like growth factor binding protein-2 (IGFBP2), circulating in the serum of patients or endogenously produced by the tumor, activates ERK1/2 and determines resistance to temozolomide by favoring the entrance into S- and G2/M-phases [28]. IL-6, either secreted by tumor or tumor-associated cells, mediates resistance to 5-fluorouracile in colon cancer cells. Upon binding the IL-6 receptor alpha (IL-6Rα)/glycoprotein 130 (GP130) complex, it triggers the activation of JAK/STAT3 and MEK/ERK1/2 pathways, which increase the anti-apoptotic proteins Bad and Bcl-2, thus preventing cell death [18]. NVP-BKM120, a pan-PI3K inhibitor, increases the IL6-dependent activity of ERK and STAT3, which render cells insensitive to the anti-proliferative effect of PI3K inhibition [115].

Activated ERK, in response to soluble factors, also mediates the acquisition of resistance during chemotherapy. CTGF, a soluble factor up-regulated by temozolomide treatment in glioblastoma, contributes to the acquisition of the resistance towards the drug by increasing the production of transforming growth factor-β1 (TGF-β1), which in turns activates the Smad/ERK1/2 pathway [116]. Dying cells damaged by chemotherapy release the high mobility group box 1 protein (HMGB1), a danger-associate molecule patterns (DAMPs) protein that binds the advanced glycation end product (RAGE) receptor and activates the down-stream ERK/Drp1 axis on residual cells, making the latter chemoresistant to a following treatment [103].

Overall, these studies suggest that cell–cell or cell–ECM dependent contacts, paracrine factors secreted by tumor-associated stromal cells, and/or endogenously produced cytokines or chemokines activate the MEK/ERK1/2 cascade in tumor cells. Working as a sensor of modifications in the extracellular environment and as a collector of different stimuli, ERK1/2 induces stress adaptation and pro-survival processes. This mechanism is valid for different stressing agents, including chemotherapy.

### 4.3. ERKs Increase Expression and Activity of ABC Transporters

Most drugs used in the studies reported above are substrates of ATP binding cassette (ABC) transporters, a family of membrane pumps effluxing chemotherapeutic drugs unrelated for structure and activity. The main transporters expressed in cancer cells include P-glycoprotein (Pgp/ABCB1), multidrug resistance related proteins (MRPs/ABCCs), and breast cancer resistance proteins (BCRP/ABCG2) [117]. Additionally, ABC transporters are involved in ERK-mediated chemoresistance. We previously demonstrated that, in resistant colon cancer, chronic lymphocytic leukemia, pleural malignant mesothelioma, non small cell lung cancer, and breast cancer, ERK1/2 phosphorylates and stabilizes the hypoxia-inducible factor-1α (HIF-1α), a strong inducer of Pgp [31,35,118,119,120]. Similarly, in hypoxic pancreatic cancer cells, ERK1/2 activates HIF-1α and up-regulates BCRP [121]. Further associations between ERK and ABC transporters were demonstrated in gastric cancers, where activated ERK1/2 induces MRP1 and BCRP [122], in acute myeloid leukemia, where ERK1/2 up-regulates Pgp and MRP1 [123], and in ovarian cancer, where the inhibition of ERK1/2 and AKT lowers the levels of BCRP [124].

It is common that ERK1/2 up-regulates ABC transporters in response to specific stimuli from the extracellular environment. In colon cancer, CD44v6, the hyaluronane receptor that is involved in cell proliferation and migration, up-regulates Pgp and BCRP, determining resistance to 5-fluorouracile and oxaliplatin [125]. CD44v6 activates RAS/MEK/ERK signaling [125]. The stimulation of ERK could represent the linkage between CD44v6 activation and Pgp/BCRP induction. The G protein-coupled receptor 55 (GPR55) activates both PI3K/AKT and the MEK/ERK1/2 axis, and promotes the nuclear translocation of HIF-1α and β-catenin, two inducers of Pgp and BCRP in pancreatic cancer cells [126]. The interaction of collagen with integrin-β1 also activates ERK1/2 signalling that up-regulates MRP1 in T-cell acute lymphoblastic leukemia, limiting the intracellular accumulation and toxicity of doxorubicin [127].

Regarding soluble factors involved in the ERK-induced up-regulation of ABC transporters, it is known that the granulocytic-colony stimulating factor (G-CSF) induces Pgp by activating the RAS/ERK axis in gastric cancer [128], and the extracellular ATP up-regulates MRP2 by increasing MEK/ERK1/2 signaling in transformed colon cells [129].

Some genetic and epigenetic mechanisms controlling the induction of ABC transporters by ERK1/2 have been identified. The soluble resistance-related calcium binding protein sorcin is often co-expressed with Pgp, since it is in the same amplicon. Interestingly, in acute myeloid leukemia cells, the elevated production of sorcin activates ERK1/2 and AKT, which up-regulates Pgp transcription [130]. The most consistent epigenetic controller of ERK is miR-20a, which down-regulates ERK-induced Pgp transcription in breast cancer cells [131] and restores chemosensitivity in osteosarcoma cells [132]. Similarly, miR-20a down-regulates Lgr1, lowering the EGFR-dependent MEK/ERK and PI3K/AKT pathways and restoring chemosensitivity in gastric cancer [133]. Moreover, the miR302 family has been identified as a down-regulator of the MAPK/ERK1/2/Pgp axis in breast cancer [134].

In stem cells, BCRP, Pgp, and ATP binding cassette subfamily B member 5 (ABCB5) are overexpressed and mediate multi-drug resistance [135,136,137,138]. Of note, specific stemness pathways activated by ERK1/2, such as the GSK3β/β-catenin pathway [24], up-regulate Pgp and ABCB5 [137,138], providing a linkage between ERK activation and chemoresistance in cancer stem cells.

Curiously, JNK and ERK1/2 are necessary to protect Pgp-expressing cells from cold stress. Inhibiting ERK1/2 in cell cultures cooled at 4 °C produces the selective lethality of resistant cells, opening new therapeutic possibilities to eradicate Pgp-expressing cells [139].

Overall, the ERK-induced chemoresistance mediated by ABC transporters is a phenomenon shared by cancers with different histological origin and genetic background and occurs either in the presence of tumor-associated stromal cells or in their absence, leading us to hypothesize that this is a robust and highly conserved mechanism. Indeed, both ERK and ABC transporters help cells to adapt to extracellular stressing conditions, activating pro-survival mechanisms. This parallel role naturally evolved into their mutual cooperation in inducing resistance to chemotherapeutic drugs.

Few reports suggest that ERK1/2 induces chemosensitization. For instance, the pan-ErbB tyrosine kinase inhibitor (TKI) dacomitinib improves the sensitivity to topotecan in BCRP-overexpressing non small cell lung cancer cells by downregulating ERK activity without any modification in BCRP expression or activity [140]. The EGFR inhibitor olmutinib increases the intracellular accumulation and cytotoxicity of doxorubicin and topotecan by binding the ATP binding site of BCRP [141]. Non small cell lung cancer cells cultured in the semi-solid medium matrigel down-regulate ERK1/2 activity and enter into a dormancy status associated with chemoresistance [142], suggesting that, under these conditions, ERK activity induces chemosensitization. This contrasting evidence is likely due to the different experimental conditions used. Additionally, the tumor molecular biology is important in determining a different response to chemotherapy. Indeed, in breast cancer cells with an oncogenic activated EGFR, ERK1/2 activity sensitizes to doxorubicin [143]. It is likely that, in these cells, the prolonged and not pulsatile ERK1/2 stimulation induces abnormal cell proliferation, as demonstrated for lung cancer cells [45]. A drug such as doxorubicin, which impairs topoisomerase II activity and is more effective in highly proliferating cells, is more effective in this situation.

Overall, ERK1/2 is generally a protective factor from chemotherapy, but the presence of different extracellular factors, different oncogenic activation of the RAS/BRAF/MEK/ERK axis, and different timing of ERK activation may sometimes result in a broad spectrum of effects, ranging from chemoresistance to chemosensitization in peculiar tumors. The molecular annotation of these additional factors may help in conceiving more personalized interventions targeting ERK activity aimed at improving chemotherapy efficacy.

## 5. ERKs are Pleiotropic Modulators of Tumor Immune-Resistance and Immune-Escape

Tumor growth is counteracted by the host immune system, particularly by the adaptive immunity compartment that includes dendritc cells (DCs), anti-tumor M1-polarized tumor-associated macrophages (TAMs), effector (helper CD4^+^ and cytotoxic CD8^+^) T-infiltrating lymphocytes (TILs), and natural killer (NK) cells. Tumor neo-antigens or DAMPs are sensed by TAMs and DCs that phagocyte cancer cells, process their antigens, and activate specific CD8^+^ T-lymphocytes clones with anti-tumor activity [144]. However, tumors can escape immune system surveillance by releasing immune-suppressive cytokines (e.g., TGF-β and IL-10) and soluble factors (e.g., kynurenine), by down-regulating major hystocompability complex (MHC) molecules, by loosing antigens, by inducing the expression of immune checkpoints (ICP) on CD8^+^ TILs (i.e., molecules that make T-lymphocytes anergic when engaged by the ICP ligands expressed on tumor cells), and by recruiting immune-suppressive cells such as T-regulatory (Treg) cells, myeloid-derived suppressor cells (MDSCs), and M2-polarized TAMs [145]. The progressive exhaustion of the effector components and the increase of immune-suppressive populations determine uncontrolled tumor growth and dissemination.

The activity of ERK, either within tumor cells or tumor-infiltrating immune cells, may modulate the tumor-immune cell interaction.

### 5.1. ERK Activity in Cancer Cell Modulates the Tumor Immune-Environment

In tumors, ERK1/2 activation has been correlated mainly with immune-suppressive properties. The effects of intratumor ERK1/2 activity are highly tumor-dependent and involve virtually all the known components determining immune-resistance or immune-escape (Figure 2).

In non small cell lung cancer, oncogenic mutated RAS increases the programmed death-1 ligand (PD-1L), i.e., the tumor-associated ligand of PD-1, an ICP widely present on CD4^+^ and CD8^+^T-lymphocytes. The increase in PD-1L/PD-1 levels induces T-cell apoptosis. This immune-escape is rescued—at least in ex vivo systems—by the combination of the anti-PD-1L Pembrolizumab and the ERK inhibitor SCH772984 [34]. In the same tumor type, in the presence of oncogenic activated EGFR and KRAS/ERK1/2, EGF promotes the secretion of IL-10 that increases the number of infiltrating Treg cells and M2-polarized TAMs [146]. Similarly, colon cancers with mutated KRAS and constitutively activated ERK1/2 secrete IL-10 and TGF-β that differentiate effectors T-cells into immune-suppressive Treg cells, favoring the immune-escape of these tumors [30]. In pleural malignant mesothelioma [35] and breast cancer [31], ERK1/2 phosphorylates STAT3 on serine 727 and activates the transcriptional induction of indoleamine 2,3 dioxygenase (IDO), the enzymes producing kynurenine. This mechanism up-regulates Treg cells and impairs the expansion of effector CD3^+^T-lymphocytes. Cytokines acting on tumor and tumor infiltrating cells may modulate the immune-environment in an ERK-dependent manner. For instance, the chemokine ligand 16 (CXCL16) reduces cell proliferation of both Treg cells and gastrointestinal sarcoma cells by down-regulating ERK1/2 activity [36]. This effect may be due to a direct anti-proliferative effect on tumor cells and/or to the elimination of immune-suppressive infiltrating Treg cells. Since MEK inhibitors and CDK4/6 inhibitors kill lung cancers and, at the same time, induce a re-population of NK cells, it has been suggested that an activated MEK/ERK axis impairs the immune-surveillance mediated by NK cells [147], further contributing to induce an immune-evasive mechanism. R-spondin, a secreted agonist of the canonical Wnt/β-catenin signaling pathway, activates ERK1/2 and STAT3 pathways upon binding to its receptor Lgr4. In Lewis lung carcinoma and melanoma, this pathway promotes the enrichment of M2-polarized TAMs and the reduction of CD8^+^TILs [37]. Similarly, CXCL2, upon binding CXCR2, activates ERK1/2, p38, and NF-kB in bladder tumor cells. These factors recruit MDSC within the tumor. The causative role of ERK has been demonstrated by the use of pharmacological inhibitors that prevent the MDSC recruitment [148]. Lactate, the end-product of the anaerobic glucose catabolism, can be released from the cancer cell and activates the ERK1/2 and STAT3 signalling in monocytes, promoting their polarization in pro-tumor M2-polarized TAMs and favoring cancer cell proliferation and migration [149]. From this perspective, highly glycolytic tumors are more prone to creating an immune-suppressive environment mediated by ERK.

ERK1/2 sometimes acts as a double-edge sword, depending on the phase of tumor progression and immune-response. Interestingly, murine mammary carcinoma, lymphoma, and fibrosarcoma with a mutated and activated RAS/RAF/MEK/ERK1/2 axis promote the expansion of activated CD8^+^TILs, producing IFN-γ that initially induces tumor cell killing. In the long term, however, this process favors the expansion of tumor clones that progressively loose antigens and escape the immune-surveillance. Oncogenically mutated clones emerge because they are favored in terms of cell proliferation compared to clones with a wild type RAS/RAF/MEK/ERK1/2 axis. Since the selected clones display a progressively increasing genetic instability, a feature that favors their escape from host immune-surveillance, the combination of immune-therapy and DNA damaging agents (including chemotherapy and radiotherapy) can be a successful strategy in removing the mutated clones [150]. The concomitant use of ERK inhibitors may be considered as an additional immune-sensitizer tool against these tumors.

In hepatocellular carcinoma, there are discordant data on the role of ERK1/2 as an immune-activating or an immune-suppressive player. One study indicates that ERK1/2 promotes an immune-suppressive environment. In immunecompetent animal models, the TKI Sorafenib that reduces ERK1/2 activity activates DCs, TAMs, TILs, and NK cells as well as reduces the expression of PD-1/PD-1L and the amount of infiltrating monocytic-MDSCs [33]. Therefore, the decreased ERK1/2 activity induced by Sorafenib attenuates the tumor-induced immune-suppression.

Contrary to the previous study, another work reports that, in patients with hepatocellular carcinoma, a high level of phosphorylated ERK1/2 within tumors is associated with an increased number of PD-1^−^ CD8^+^ T-lymphocytes and with a higher overall survival rate compared to patients with low phosphorylated ERK1/2 [151]. This work suggests that a higher activity of ERK1/2 within hepatocellular carcinoma cells is a useful marker of a pro-inflammatory and an immune-active/anti-tumor environment. The discrepancies between these two studies may suggest that, at least for hepatocellular carcinoma, ERK activation has a different impact on the immune-infiltrate depending on the species analyzed. Since the immune-reactivities of mice and humans are different, caution is required when the results of studies obtained in murine models are extrapolated to humans.

With the intense use of ICP inhibitors as approved treatments in different tumors, the role of ERK in controlling the expression of ICP and respective ligands has become of great interest. Multiple mutual loops between ERK and ICP ligands have been identified. ERK1/2-dependent pathways up-regulate ICP ligands, such as lymphocyte-activation gene 3 (LAG-3) in melanoma [152], and PD-1L in malignant pleural mesothelioma [153], non small cell lung cancer [154], breast cancer [155], prostate cancer [32], bladder cancer, and multiple myeloma [156].

Activating oncogenic mutations in EGFR or RAS up-regulates PD-1L at a transcriptional level by activating MEK/ERK1/2 cascade [34,157]. Depending on the tumor type, the RAS/MEK/ERK pathway acts alone or cooperates with other oncogenic pathways [e.g., JNK-, PI3K/AKT-, IL-6/STAT3, and echinoderm microtubule associated protein-like 4/anaplastic lymphoma kinase (EML4/ALK)-dependent pathways] in increasing PD-1L [124,158,159,160,161,162]. Additionally, post-translational mechanisms are responsible for the increase of PD-1L mediated by ERK1/2. Indeed, in lung and colon cancers with oncogenic activated RAS, ERK1/2 phosphorylates the protein tristetraprolin that binds the 3′UTR of PD-1L mRNA, increasing its stabilization. As a consequence of the increased levels of PD-1L on cancer cells, a low intratumor CD8^+^/Treg cells ratio and a low percentage of CD8^+^INFγ^+^ cells are detected [163]. In non small cell lung cancer, the H-RAS/MEK/ERK1/2 pathway not only up-regulates PD-1L but also down-regulates the NK group 2 member D (NKG2D), a key ligand activating NK cells [164]. This double action produces a strong immune-suppression. In the vast majority of the studies reported, the proof of concept of ERK involvement derives from the abrogation of PD-1L up-regulation when cells are treated with pharmacological inhibitors of ERK.

On the other hand, ERK is a downstream transducer of specific ICP ligands in tumor cells. For instance, in glioblastoma, PD-1L determines EMT and invasion by up-regulating the H-RAS/MEK/ERK1/2 pathway [38]. Similarly, the interaction of LAG-3, expressed on T-lymphocytes, with MHC molecules present on melanoma cells, activates ERK1/2, promoting both increased melanoma cell proliferation and immune-escape [152]. Cytotoxic T-Lymphocyte antigen 4 (CTLA-4), another ICP expressed in non small cell lung cancer, activates ERK1/2 and TGF-β [165]. The ERK-mediated increased proliferation and the TGF-β-mediated induced immune-suppression both contribute to promote tumor growth and immune-escape.

### 5.2. ERK Activity in Tumor/Infiltrating Immune Cells

In immune cells, ERK1/2 is a general pro-survival factor. Therefore, its activity may have immune-activating or immune-suppressive results depending on the cell types infiltrating the tumor (Figure 3).

For instance, ERK1/2 activates the proliferation of CD8^+^ TILs and stimulates the secretion of cytokines and lytic granules [166], enhancing the cytotoxic activity. In follicular lymphoma, CD8^+^ T-cells expressing the ICP T-cell immunoglobulin mucin 3 (TIM-3) have a defective ERK1/2 signaling, low proliferation, and degranulation [167]. This study does not clarify if ERK is the down-stream effector of TIM-3 or the cause of the anergic lymphocytic phenotype of T-cells, but it demonstrates that a lower activity of ERK determines a poor cytotoxic activity upon T-cell receptor (TCR) engagement. The prolonged engagement of TIM-3, however, exhausts CD8^+^ TILs and down-regulates their pro-survival pathways, including ERK1/2 activity. This make TILs anergic and more prone to undergoing spontaneous apoptosis [167]. In Jurkat T-cells, ERK1/2 is a transcriptional activator of TIM-3 [168], indirectly suggesting that ERK is likely upstream TIM-3 in follicular lymphoma cells as well.

In highly immune-evasive tumors such as kidney cancer, infiltrating CD8^+^ TILs have a constitutively low level of ERK phosphorylation. The low activity of ERK1/2 is due to the inhibitory phosphorylation exerted by the diacylglycerol kinase-α (DGK-α) that is expressed more in TILs compared with non-TILs. The inhibition of DGK-α with low dose of IL-2 restores ERK1/2 activity in CD8^+^TILs, as well as their proliferation and cytotoxic activity [169].

The oncogenic hepatitis B surface antigen (HBsAg) promotes the differentiation and the expansion of monocytic-MDSCs by activating ERK1/2 and the IL-6/STAT3 axis. Consequently, effector T-cells proliferation is reduced [170]. These data suggest that, in patients with chronic hepatitis induced by hepatitis B virus, an active ERK1/2 may create a tumor-developing environment. ERK1/2 together with STAT3 and AKT also promotes the functional proliferation and activity of MDSCs in breast cancer, where the three pathways are down-regulated by the onco-suppressor aminoacyl-tRNA synthetase-interacting multifunctional protein 1 (AIMP1) [171]. Although ERK1/2 cooperates with Src, AKT, and protein kinase C (PKC) in promoting MDSC differentiation, ERK1/2 is the most potent driver of this process, because the selective ERK inhibitor VTX-11e is the most potent agent preventing the differentiation of MDSCs [172].

In T-helper 17 (Th17) infiltrating cells, a multifaceted T-cell subset with either anti-tumor or pro-tumor activity, ERK1/2 and NF-kB increase the recruitment of immature granulocytic-MDSCs, which create a local immune-suppressive and VEGF-independent pro-angiogenic environment [173]. In this case, ERK plays the dual role of inducing immune-suppression and resistance to anti-VEGF antibodies. Intriguingly, ERK1/2 is a fine modulator of Th17 activity. On the one hand, it is necessary to increase the development of Th17 cells with pro-inflammatory functions; on the other hand, in the presence of an immune milieu that favors the Th17 differentiation, the inhibition of ERK1/2 promotes the acquisition of a Treg phenotype with immune-suppressive functions [174]. Therein, ERK activity in combination with microenvironment related factors may represent the turning point to shift the same T-cell subset from an anti-tumor towards a pro-tumor population.

The tumor metabolic profile also contributes to create anti-tumor or pro-tumor immune activity. For instance, lactate produced by highly glycolytic tumors activates the ERK1/2 and the STAT3 signaling in monocytes, promotes their polarization in pro-tumor M2 TAMs, and favors tumor cell proliferation and migration [149]. These findings suggest that tumors with a more anaerobic metabolism are more prone to creating an immune-suppressive environment, as they are more prone to be chemoresistant [101,102] using ERK-dependent mechanisms.

While the activation of ERK1/2 in tumor cells mainly induces immune-resistance and immune-escape, the physiological activation of ERK in immune populations leads to their proliferation, resulting in pro-tumor or anti-tumor effects depending on the type of population expanded. Therein, ERK1/2 inhibition may paradoxically produce an immune-suppressive environment if it reduces the expansion of anti-tumor immune-populations. The continuous interaction between tumor and immune cells modulates ERK1/2 in the immune system via specific co-receptors or soluble factors released by the tumor, increasing the ratio between immune-suppressive/pro-tumor and immune-active/anti-tumor cells. The disruption of these vicious circles may attenuate the ERK-induced immune-suppression in the most immune-evasive tumors.

## 6. ERK-Driven Circuitries Determining Chemoresistance and Immune-Resistance

ERK is at the cross-road of different circuitries determining chemoresistance and immune-resistance. Of note, some of these circuitries are druggable and may induce chemo- and immune-sensitization (Figure 4).

We previously demonstrated that, in chemoresistant colon cancer, breast cancer, and malignant pleural mesothelioma, RAS/ERK1/2 inhibition reduces the phosphorylation and the stability of HIF1α, a transcriptional inducer of Pgp, even in normoxia. The reduction of Pgp expression allows an increased intracellular accumulation of different chemotherapeutics, including doxorubicin [31,35,118]. Besides killing tumor cells by inhibiting topoisomerase II, inducing oxidative and nitrosative stress, or altering mitochondrial functions, doxorubicin is able to induce an immunogenic cell death (ICD), i.e., a type of death that causes tumor cells to be recognized by the immune system [144]. The higher the intracellular content of doxorubicin is, the higher its ability to promote the release of DAMPs (ATP and HMGB1) will be. These activate local DCs and induce the translocation of calreticulin, a calcium-sensor residing within endoplasmic reticulum onto the tumor cell surface. Here, calreticulin acts as an “eat me signal”, further enhancing the tumor cell phagocytosis by DCs and the consequent expansion of CD8^+^T-cells with anti-tumor activity [144].

Pgp counteracts ICD by at least two mechanisms. First, it effluxes doxorubicin. Second, when Pgp is abundant on the tumor cell plasma-membrane, it impairs the immune-activating functions of calreticulin, hampering the tumor cell phagocytosis mediated by DCs [175]. Thus, inhibiting the RAS/ERK1/2/HIF1α/Pgp axis promotes ICD by increasing the intracellular retention of doxorubicin and restoring the immune-activating functions of calreticulin [31,35,119].

Moreover, multi-drug resistant cells of malignant pleural mesothelioma [35] and breast cancer [31] have a constitutively activated ERK1/2/STAT3 pathway that induces the transcription of IDO. This enzyme catabolizes tryptophan, an essential aminoacid for effector T-cells, and produces the immune-suppressive metabolite kynurenine. Inhibiting the ERK1/2/STAT3 axis reduces Treg cells expansion and re-instates a correct proliferation of effector TILs [31,35], turning an immune-suppressive/pro-tumor environment into an immune-active/anti-tumor one.

Different approaches targeting the RAS/ERK1/2/HIF1α/Pgp axis have been experienced in order to induce sensitization of chemo-immune-resistant cells. Besides ERK pharmacological inhibitors such as PD98059 [119], inhibitors of farnesyl pyrophosphate synthase (FPPS) indirectly reduce RAS/ERK1/2 activity by limiting the supply of FPP to RAS [118]. A potent FPPS inhibitor is the aminobisphosphonate zoledronic acid, a Food and Drug Administration (FDA)-approved drug for osteoporosis and metastatic tumors. Using self-assembled nanoparticles carrying zoledronic acid that have a high intratumor delivery of the drug by preventing the bone uptake, we contemporarily induced chemo- and immune-sensitization of resistant pleural malignant mesothelioma [35] and breast cancer [31] by reducing the activity of RAS/ERK1/2/HIF1α/Pgp and RAS/ERK1/2/STAT3/IDO axes.

Besides chemotherapeutic drugs such as doxorubicin, TKI may activate an anti-tumor immune response. For instance, axitinib, which down-regulates PDGFR/ERK1/2 signaling, is more effective in immunecompetent mice bearing orthotopic glioblastoma than in immunedeficient animals [176]. This work indirectly suggests that ERK suppression mediates the efficacy of axitinib by engaging the immune system. Therein, glioblastoma with a constitutively activated ERK, e.g., the glioblastoma characterized by EGFR/PDGFR amplification or oncogenic mutations, may result as both resistant to targeted-therapy and immune-resistant.

Immune-therapy based on the ICP/ICP ligands blockade is often less effective in chemoresistant tumors. The high activity of ERK1/2 in resistant cancers increases the expression of ICP-L, leading us to hypothesize that higher doses of ICP/ICP ligands inhibitors are required against chemoresistant tumors. For instance, in non small cell lung cancer with oncogenic activated EGFR, the activation of MEK/ERK1/2 in combination with Smad3 and mTOR promotes resistance to cisplatin, induces the EMT program, and up-regulates PD-1L. The combined inhibition of MEK/ERK, Smad3, and mTOR rescues the sensitivity to cisplatin and reduces the expression of PD-1L [177]. This could be an efficient strategy for improving the efficacy of endogenous effector T-cells and of anti-PD-1/PD-1L immune-therapy. MEK/ERK is not the only driver of PD-1L expression; for instance, it cooperates with PI3K/AKT in non small cell lung cancer bearing EML4/ALK fusion [178]. Once again, the presence of other oncogenic pathways determines to which extent ERK1/2 up-regulates PD-1L expression and ERK inhibitors are effective in combination with anti-PD-1/PD-1L inhibitors.

Cisplatin-resistant non small cell lung cancer cells have higher expression of PD-1L and lower expression of NKG2D ligands compared to their chemosensitive counterpart. Consequently, NK cells are poorly activated within the tumor, while anergic PD-1^+^ CD8^+^ T-lymphocytes are abundant. Intriguingly, the ERK inhibitor U0126 rescues the efficacy of anti-PD-1L immune-therapy [179]. Since ERK also controls chemoresistance in non small cell lung cancer [120], further strategies with possible combinations of ERK inhibitors, PD-1/PD-1L inhibitors, and chemotherapy are noteworthy to be tested in preclinical models of this resistant tumor.

Moreover, PD-1L activates MAPK/ERK and PI3K/AKT signaling, increases Pgp expression, and determines resistance to doxorubicin in breast cancer [180]. The presence of PD-1 in TILs induces resistance to the anti PD-1/PD-1L antibodies and engages PD-1L present on the surface of breast and prostate cancer cells. Upon this engagement, PD-1L activates the ERK1/2 and the mTOR pro-survival pathways, determining resistance to doxorubicin and docetaxel [181]. Furthermore, in these tumors, the use of ERK inhibitors may improve the efficacy of chemotherapy and anti-PD-1/PD-1L-based immune-therapy.

Of note, chemotherapy itself can select chemo-and immune-resistant clones. In head and neck cancers, the treatment with cisplatin increases the expression of PD-1L and the activity of MAPK/ERK1/2 [182]. Although no data on patient response and survival are available, it is likely that cisplatin, by expanding PD-1L cells, contributes to select clones with an increasing immune-suppressive potential. These clones are likely poorly sensitive to cisplatin because they have a high activity of ERK1/2. Similarly, doxorubicin-treated lymphoma cells progressively increase PD-1L on bone marrow stromal cells, producing a functional anergy of effector T-cells [183]. This is the first experimental evidence of a “chemotherapy-induced” chemo-immune-resistance.

If, in most cases, the cross-talks determining immune-resistance arise from the higher activity of ERK1/2 within tumor cells, in some tumors, these cross-talks are triggered by the immune-infiltrating cells. In triple negative breast cancer, IL-17A produced by Th17 TILs activates the MAPK/ERK1/2 pathway within the tumor and induces resistance to docetaxel by up-regulating anti-apoptotic Bcl-2 family proteins [184]. Since Th17 may exert a strong immune-suppressive activity, these tumors have a chemo-immune-resistant phenotype triggered by the immune-infiltrate and not by the tumor. TAMs induce the progressive loss of sensitivity towards taxol in breast cancers by activating the MEK/ERK1/2 pathway that accelerates mitosis and reduces the ability of taxol to interfere with mitotic spindle. Either MEK/ERK inhibitors or small synthetic molecules depleting TAMs restore taxol efficacy [185], highlighting another mechanism of immune system-induced chemoresistance. Since the chemosensitivity is obtained by BLZ945, a TAM-depleting agent that inhibits the colony stimulating factor-1 receptor (CSF-1R), we may hypothesize that CSF-1 could be involved in the TAM-induced chemoresistance.

The existence of multiple cross-talks determining chemo-immune-resistance—mostly regulated by cytokines, soluble factors, or surface receptors present either on tumor or immune cells—may open new possibilities of intervention based on recombinant soluble cytokine receptors, inhibitory small molecules, or monoclonal blocking antibodies that may have an additive effect to ERK inhibitors in disrupting the chemo-immune-resistance loops.

## 7. Conclusions and Future Perspectives

Most tumors have redundant pathways allowing their survival. Therein, inhibiting TKRs with specific TKIs or inhibiting their immediate downstream effectors may result in the compensatory up-regulation of parallel oncogenic pathways or downstream oncogenic transducers. A well-known case is the inhibition of BRAF that may produce the aberrant activation of its downstream transducer MEK, selecting clones resistant to the initial targeted therapy. ERK1/2 is one of the last transducers of the cascade RTK/RAS/BRAF/MEK, and it is at the cross-road of many cellular pathways controlling cell cycle, apoptosis, chemoresistance, immune-resistance, and immune-escape. Since ERK is a sensor of multiple extracellular signals and a transducer common to several RTKs, it may be regarded as an Achille’s heel in tumors. Several ERK1/2 inhibitors have been developed, and some of them reached phase I clinical trials in solid and hematological tumors [79].

The therapeutic success of ERK1/2 inhibitors is, however, below expectations due to multiple limitations. First, the type of oncogenic activation, e.g., a tonic and continuous stimulation instead of a temporally limited and pulsatile stimulation, widely affects the efficacy of ERK inhibition, producing responses ranging from an “expected” anti-proliferative effect to a paradoxical pro-proliferative effect [17]. A deep knowledge of the oncogenic mutations of the RAS/BRAF/MEK/ERK1/2 pathways is needed to refine the precision medicine based on ERK inhibition.

Second, as it occurs for BRAF and MEK inhibitors, resistance to ERK1/2 inhibitors due to ERKs mutations or amplification has been documented [79]. Such resistance is expected to increase with the increasing use of ERK inhibitors. Although the resistance can be overcome by combining different inhibitors of ERK1/2 pathways, it dampens the enthusiasm around ERK1/2 inhibitors as anti-tumor agents alone.

Third, ERK1/2 controls the proliferation and the differentiation of non-transformed cells. Hence, its inhibition may have severe consequences on the physiology of normal tissues.

If ERK inhibitors alone have pros and cons as anti-tumor agents, the emergent role of ERK1/2 in controlling the response to chemotherapy, to the anti-tumor activity of the host immune system, and to the immune-therapy opened new perspectives on the possible use of ERK1/2 inhibitors as chemo- and immune-sensitizers. The main issue from this perspective is achieving a good selectivity. Indeed, inhibiting ERK1/2 in tumor cells generally reverses chemoresistance and tumor-induced immune-suppression. Inhibiting ERK1/2 in the tumor-infiltrating cells is a high-risk task, because ERK1/2 inhibitors can reduce the expansion of both effectors and immune-suppressive cells. Increasing the selectivity of ERK inhbitors towards specific populations of the tumor environment, e.g., by conjugating ERK inhibitors with ligands or monoclonal antibodies recognizing specific antigens on tumors or immune cells, may overcome this lack of specificity. The technical feasibility and the preclinical efficacy of these synthetic conjugates must still be verified. Moreover, given the high intra-tumor and inter-patients variability in the immune-infiltrate, each patient may present qualitative and quantitative differences in the intra-tumor immune-infiltrate. Only a routinely performed immune-phenotyping of each tumor sample can help to decide if ERK1/2 inhibitors are useful in specific patient groups.

An exploitable feature is the presence of multiple ERK-driven circuitries determining chemoresistance, immune-resistance, and immune-escape. In aggressive tumors, the disruption of these vicious circuitries with blockers of ERK activity, including ERK pharmacological inhibitors and metabolic modifiers reducing RAS/ERK activation (such as aminobisphosphonates) or TKIs, may enhance the efficacy of chemotherapy or immune-therapy. Choosing the right combination of drugs will be possible only after extensive preclinical investigations on patient-derived xenografts and patient-derived immune-organoids that reproduce the tumor biology and the tumor-stroma-immune cell interactions present in the real tumors in the closest way. This will be the next step towards a precision medicine based on ERK inhibitors as effective chemo-immune-sensitizers.

## Figures and Tables

**Figure 1 ijms-20-02505-f001:**
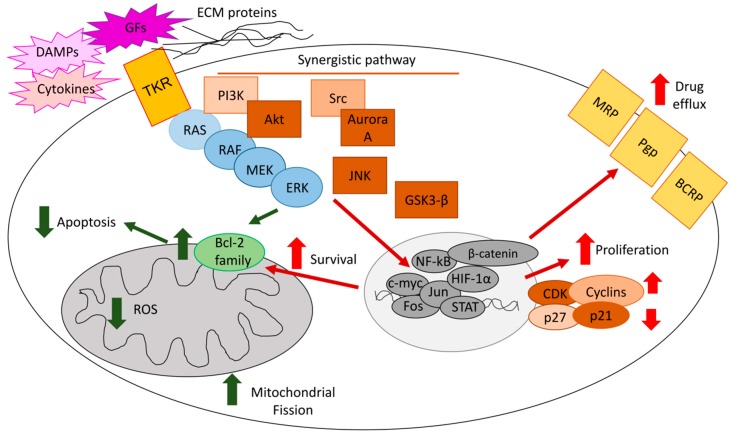
Mechanisms of extracellular signal-regulated kinase (ERK)-induced chemoresistance. The RAS/RAF/MEK/ERK axis [rat sarcoma 2 viral oncogene homolog/rapidly accelerated fibrosarcoma/mitogen-activated protein kinases (MAPK)-extracelluar signal-regulated kinase], upon the activation of tyrosine kinase receptors (TKR) by growth factors (GFs), cytokines, or cell–extracellular matrix (ECM) interaction, cooperates with phosphoinositide 3-kinase (PI3K)/protein kinase B (AKT), c-Jun amino-terminal kinase (JNK), steroid receptor coactivator-1 (Src)/Aurora kinase, glycogen synthase kinase 3-β (GSK-3β)/β-catenin signals, activating a pool of transcription factors that determine: (i) increased cell proliferation by up-regulating cyclins and cyclin-dependent kinase (CDKs) and/or down-regulating the cell cycle check-points p21 and p27; (ii) prevention of apoptosis by decreasing Bcl-2 proteins, reducing mitochondrial reactive oxygen species (ROS) and mitochondrial fission; (iii) increased expression and activity of drug efflux transporters, such as P-glycoprotein (Pgp), multidrug resistance related proteins (MRPs), and breast cancer resistance proteins (BCRP). All these mechanisms make a tumor cell more resistant to chemotherapy. Green arrows: apotosis-related mechanisms; red arrow: pro-survival-related mechanisms.

**Figure 2 ijms-20-02505-f002:**
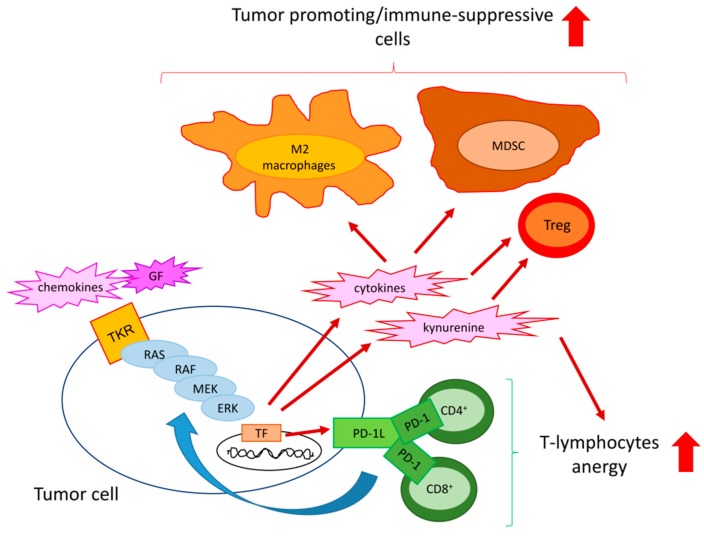
Mechanisms of ERK-induced immune-resistance or immune-escape. In response to chemokines and growth factors (GFs), specific tyrosine kinase receptors (TKR) activate the RAS/RAF/MEK/ERK axis and the down-stream transcription factors (TF) in tumor cells. As a result, cancer cells produce immune-suppressive cytokines [e.g., interleukin (IL)-10 and transforming growth factor-β (TGF-β)] and metabolites (e.g., kynurenine) that reduce the proliferation of effector CD4^+^ and CD8^+^T-lymphocytes and expand tumor-promoting/immune-suppressive cells, such as M2-polarized macrophages, myeloid derived suppressor cells (MDSC), and T-regulatory (Treg) cells. The RAS/RAF/MEK/ERK axis also up-regulates the immune-checkpoint ligand programmed death-1 ligand (PD-1L), which expands PD-1 rich/anergic CD4^+^ and CD8^+^ T-cells and fuels ERK activity with a positive feedback mechanism. Exploiting these multiple strategies, ERK activation in tumor cells contributes to create a tumor-promoting/immune-suppressive environment.

**Figure 3 ijms-20-02505-f003:**
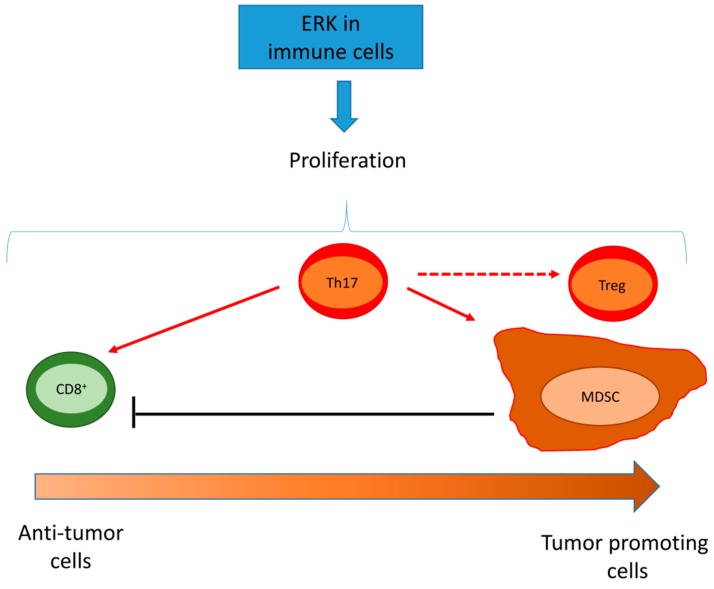
Effects of ERK activation in tumor infiltrating cells. ERK induces the proliferation of anti-tumor immune cells, such as CD8^+^ T-lymphocytes, immune-suppressive cells, such as myeloid derived suppressor cells (MDSC) and T-regulatory (Treg) cells, and cells that have either pro-tumor or anti-tumor activities, such as T-helper 17 (Th17) cells. Th17 cells can promote the pro-inflammatory phenotype of CD8^+^T-lymphocytes, differentiate into cells with a Treg phenotype, and expand MDSCs that induce apoptosis of CD8^+^ T-cells. Depending on the type of immune-population characterized by high ERK activity, a wide range of effects (ranging from an anti-tumor to a tumor tolerant/immune-suppressive environment) is obtained. Dashed arrows indicate a direct activation by Th17 cells; the dotted arrow indicates the product of a differentiation of Th17 cells.

**Figure 4 ijms-20-02505-f004:**
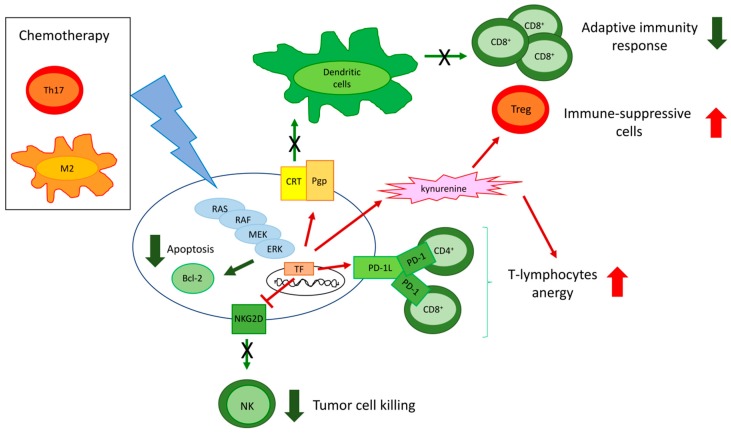
Cross-talks between tumor and immune cells determining chemo-immune-resistance. Chemotherapy and immune cells, e.g., T-helper 17 (Th17) cells producing IL-17 and M2-polarized macrophages producing colony stimulating factor-1, activate the RAS/RAF/MEK/ERK pathway in tumor cells, leading to the up-regulation of the Bcl-2 proteins that promote chemoresistance by reducing apoptosis. Moreover, in response to chemotherapy, resistant tumors activate the RAS/RAF/MEK/ERK axis that: (i) up-regulates P-glycoprotein (Pgp), a drug efflux transporter and an inhibitor of the calreticulin (CRT)-triggered phagocytosis by dendritic cells (DCs), avoiding the establishment of a durable adaptive immune response mediated by CD8^+^ T-cells; (ii) promotes the secretion of kynurenine that expands the immune-suppressive T-regulatory (Treg) cells; (iii) up-regulates the immune-checkpoint ligand programmed death-1 ligand (PD-1L) that expands PD-1 rich/anergic CD4^+^ and CD8^+^ T-cells; (iv) down-regulates the surface protein NKG2D, preventing the cytotoxic activity of natural killer (NK) cells. Chemo-immune-resistance is sustained by either chemotherapy or immune cell-induced mechanisms, all converging on an increased ERK activation. Green arrows: decreased pathways (thin arrows) and events (thick arrows); red arrow: increased pathways (thin arrows) and events (thick arrows).

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
