# Peer review of "ERK is a Pivotal Player of Chemo-Immune-Resistance in Cancer"

_ijms, 2019, doi:10.3390/ijms20102505_

Round 1
Reviewer 1 Report
The review covers an important topic, however, the presence of multiple grammatical errors, misspellings, and incorrect word usage throughout the manuscript dramatically reduces my enthusiasm for this manuscript. I also think the first part of the manuscript could be organized in a more logical and easy to follow manner (the immune-resistance and immune-escape section and the sections following that are much easier to follow). The figures are well done and informative. Specific concerns are outlined below.
1. There are numerous examples of grammatical errors, misspellings, and incorrect word usage throughout the manuscript. This is distracting and needs to be addressed. Examples include;
Line 10: I suggest changing ‘actors’ to ‘molecules’ (avoid having the words ‘act’ and ‘actors’ in one sentence)
Line 13: should say ‘cells’ not ‘cell)
Line 16: I suggest changing ‘immune-system’ to immune system responses’
Line 19: should say ‘cons’ not ‘cos’
Line 25: should say ‘so called’ not ‘socalled’
Lines 33 and 35: I don’t understand why italics are used here
Line 57: ‘Experimented’ should be changed to ‘experienced’
Line 67: I assume you mean to same ‘cell fate’ not ‘cell faith’
There are many more examples, but it would take me too long to make note of all of these
2. To help frame the paper, it would be helpful to make note of which cancer types are most frequently associated with Erk family member-mediated chemo-immune resistance earlier in the manuscript (perhaps at the end of the introduction or at the beginning of section 2).
3. The author should be careful with their use of the term ‘up-regulate’. In some instances it would be more appropriate to use the term ‘activate’ because the expression level of molecules is not affected, simply their activation status.
4. There should be more discussion of Erk1/2 mutations and amplifications; their frequency in various cancers and their contribution to chemo-immune resistance should be noted
Author Response
Reply to Reviewer 1
The review covers an important topic, however, the presence of multiple grammatical errors, misspellings, and incorrect word usage throughout the manuscript dramatically reduces my enthusiasm for this manuscript. I also think the first part of the manuscript could be organized in a more logical and easy to follow manner (the immune-resistance and immune-escape section and the sections following that are much easier to follow). The figures are well done and informative. Specific concerns are outlined below.
1). There are numerous examples of grammatical errors, misspellings, and incorrect word usage throughout the manuscript. This is distracting and needs to be addressed. Examples include;
Line 10: I suggest changing ‘actors’ to ‘molecules’ (avoid having the words ‘act’ and ‘actors’ in one sentence)
Line 13: should say ‘cells’ not ‘cell)
Line 16: I suggest changing ‘immune-system’ to immune system responses’
Line 19: should say ‘cons’ not ‘cos’
Line 25: should say ‘so called’ not ‘socalled’
Lines 33 and 35: I don’t understand why italics are used here
Line 57: ‘Experimented’ should be changed to ‘experienced’
Line 67: I assume you mean to same ‘cell fate’ not ‘cell faith’There are many more examples, but it would take me too long to make note of all of these
We carefully checked the manuscript and corrected grammatical errors, misspelling and incorrect words.
2) To help frame the paper, it would be helpful to make note of which cancer types are most frequently associated with Erk family member-mediated chemo-immune resistance earlier in the manuscript (perhaps at the end of the introduction or at the beginning of section 2).
We listed the main tumors where ERK is involved in chemoresistance and immune-resistance/immune-escape at the end of the Introduction (page 2, line 51).
3) The author should be careful with their use of the term ‘up-regulate’. In some instances it would be more appropriate to use the term ‘activate’ because the expression level of molecules is not affected, simply their activation status.
We checked the use of “up-regulate”. We changed it into “activate” where only the activity and not the expression of the target protein is modified.
4) There should be more discussion of Erk1/2 mutations and amplifications; their frequency in various cancers and their contribution to chemo-immune resistance should be noted
We thanks the Reviewer for the useful comment. We discuss the frequency of mutations in RAS/RAF/MEK axis, and the frequency of ERK1/2 mutations and amplifications at page 2, line 67.
We reported that oncogenic mutations in up-stream activators are common in tumors and determine the constitutive activation of ERKs. For instance, RAS constitutively activating mutations occur in 90% pancreas cancers, 50% colon and thyroid cancers, 30% lung cancers, 25% melanomas [39]. Oncogenic missense mutations of RAS, concentrated in the specific hot spots around glycine at position 12 and glutamine at position 16, have been detected in 30% of all cancers [39]. The downstream effectors B-RAF is mutated in 7% of all cancers, ranging from 100% of hairy cell leukemias to 50-60% melanomas, 40-60% thyroid cancers, 5-10% colorectal cancers [40]. V600E missense mutation is by far the most common mutation [40, 41]. MEK mutations have been detected in melanomas [42], ovarian cancers [43] and gliomas [44]. The presence of oncogenic mutations in the components of RAS/RAF/MEK axis determines an abnormally stable and prolonged activation of ERK, instead of its physiological pulsatile activity. Such prolonged activation induces the constitutive transcription of the pro-proliferative genes c-jun, EGR1 and cyclin D1 [45].
According to the Tissue Cancer Genome Atlas (TCGA) available datasets, ERK1 is amplified in 16% ovary cancers, 9% bladder cancers, 6% in lung and breast cancers, with lower percentage in the other cancer types. The percentage of mutations is always below 1%, with prevalent synonymous or missense mutations that have moderate or low impact, inducing no change or low reduction in the kinase activity (https://portal.gdc.cancer.gov/genes/ENSG00000102882). The amplifications of ERK2 are present mainly in ovary, bladder, lung and breast cancers with percentage of 21%, 13%, 12% and 5% cases, respectively. No more than 2% of all cancer types have missense mutations or deletions in the 3’-untranslated region (UTR) of ERK2, with moderate impact or probably deleterious meaning on the protein activity (https://portal.gdc.cancer.gov/genes/ENSG00000100030). Such low percentage of deleterious mutations is not surprising: since ERK1/2 is a pro-survival factor, it is expected that cancer cells with deleterious mutations are eliminated, while the clones most adapted to survive are maintained. Surprisingly, few activating mutations have been detected in ERK1 and ERK2.
Since many activating mutations are contained in the upstream molecules (such as RAS, BRAF and MEK), ERKs activating mutations or amplifications are not so crucial to maintain cell proliferation and survival. More than activating mutations or amplifications in ERK1/2, the abnormal activation of these kinases in consequence of the oncogenic mutations in RAS/RAF/MEK promotes cell survival and the adaptation to unfavorable conditions, including chemotherapy and immune-system anti-tumor response.
We highlighted when an oncogenic mutations of RAS/RAF/MEK/ERK1/2 axis was involved in chemoresistance and immune-resistance throughout the revised manuscript.
Reviewer 2 Report
Somehow, I can not leave my comment in this box.
Please find attached PDF file of my comment for this manuscript.

Author Response
Reply to Reviewer 2
This review manuscript focuses on the role of ERK for the resistant mechanisms against cyto-toxic chemotherapy and immune-therapies and the potential of targeting ERK in cancer treatments.
The manuscript covers a broad range of aspects of ERK in diverse cancer types and immune cells in tumor microenvironment, in the relationships with upstream and downstream pathways, with citations of a large number of references mostly published recently, thus provides important updated information regarding one of the most popular topics in cancer research field.
However, it seems that the manuscript is still in preliminary shape, as it harbors issues; numerous mistakes, missing relevant facts, a room to improve the structures of the topics and clarify the sentences to be comprehensive to general readers.
Here I tried to line up some but not all of the points to be addressed.
Following the advices of the Reviewer, we in depth re-analyzed the manuscript, corrected conceptual issues and grammatical errors, implemented each section by adding and discussed the relevant points indicated by the Reviewer. We revised the English style in order to make the sentences clearer for the general readers.
1) 64 -- ERKs are activated in response to extracellular stimulus
– not only RTK, integrins and cytokines are major mediators of the extra-cellular input into the RAS/ERK signaling axis associated with chemotherapy resistance.
Just for integrin/ERK in chemo-resistance, for example, there is abundant evidence in the literature like below,
https://www.ncbi.nlm.nih.gov/pubmed/25572304
https://www.hindawi.com/journals/cherp/2012/283181/
We introduced the role of integrins and cytokines in activating ERK1/2 in section 2 (page 3, line 104)..
We implemented the already cited literature with additional original works highlighting the mechanisms by which integrins, cytokines and chemokines, or their cross-talk, mediate chemoresistance in sections 4.2 and 4.3 of the revised manuscript (page 7, line 293 and 312; pag e9, line 360).
2) 91 –In contrast with the high number of activating pathways, few endogenous
antagonists of ERK are known: among this,…
In this section, several negative regulators are missing, including the major negative regulators of RAS/MAPK axis, such as NF1 (tumor suppressor RAS-GAP;GTPase activating protein) and RASA1 (another RasGAP). Also, another negative regulators that directly antagonize MAPK are ERK-specific phosphatases (DUSP4, DUSP6).
https://www.ncbi.nlm.nih.gov/pmc/articles/PMC5669868/
https://www.nature.com/articles/nature25187
http://clincancerres.aacrjournals.org/content/clincanres/early/2018/01/17/1078-0
Since the focus of the review was not on the mechanism of ERK activation or inhibition, but on its role in chemo- and immune-resistance, section 1-3 were rather synthetic in the first version.
We recognize that that we omitted specific key points, We implemented these sections according to the Reviewer’s indications in the revised version.
We included a paragraph on RAS/ERK inhibitors at page 4, line 137.
3) 122- senescence vy impairing à by impairing
We corrected the typo..
4) 135- For instance, ERK1/2 protect from oxidative stress-induced damage Chinese hamster lung fibroblasts [42] and neuronal cells where ERK 1/2 is activated by deglycase protein DJ-1 [43]. à ERK1/2 protect Chinese hamster lung fibroblasts from oxidative stress-induced damage [42]. This sentence is not clear. Both 42 and 43 are related to DJ-1?
We modified the sentence. We separated the original sentence in two parts, since only reference [43] is focused on DJ-1 (page 5, line 187).
5) 138 – activating ETK à RTK
We corrected the typo.
6) 144 -- Although ERK1/2 countercats à counteracts
We corrected the typo.
7) 181 -- and breast cancers [53] MEK/ERK1/2 confers a multi-drug resistant phenotype
We corrected the typo.
8) 360 --The activity of ERK, either within tumor cells or tumor-infiltrating immune cells, may modulate the otumor-immune cell interaction. à tumor-immune
We corrected the typos.
9) 417-- In hepatocellular carcinoma,
-- In this paragraph, it is not clear in ref 116 for the contribution of ERK in
sorafenib treatment effects.
--ref 117 mostly shows correlation of phospho-ERK and active CD8 T-cells, thus
suggesting that p-ERK might work as a useful marker
We specified that the main conclusion of ref [116] (ref [33] in the revised version) is that a reduced activity of ERK in mice hepatocellular carcinomas, promoted by Sorafenib, is associated to an increased ratio between effector cells and exhausted/immune-suppressive cells. Therefore, from this study it seems that the Sorafenib-mediated decrease in ERK activity attenuates the tumor induced immune-suppression. Contrarily to the previous study, another work reports that in patients with hepatocellular carcinoma phosphorylated ERK1/2 in tumors is associated with an increased number of PD-1- CD8+ T-lymphocytes and a higher overall survival than in patients with low phosphorylation of ERK1/2 [177; ref. 151 in the revised work]. This work suggests that activated ERK within hepatocellular carcinoma cells is a useful marker of a pro-inflammatory and immune-active/anti-tumor environment. The discrepancies between these two studies may suggest that at least for hepatocellular carcinoma ERK activation may have a different impact on the immune-infiltrate, depending on the species analyzed. Since the immune-reactivity of mice and humans are often different, caution is needed when the results of studies obtained in murine models are extrapolated to humans (page 12, line 145).
10) The role of ERK in regulation of PD1/PD-L1 expression is important. There are papers initially demonstrating but missing in the manuscript. For example, https://www.sciencedirect.com/science/article/pii/S1074761317305174?via%3Dihub
Regarding PD-1/PD-L1 regulation by MAPK, in this recent review, there is a section mentioning about MEK/ERK’s role on PD-L1 regulation in cancer (P.443~). It would be useful to see the consistency for the selection of papers from the literature.
https://www.cell.com/immunity/comments/S1074-7613(18)30090-6
We revised the consistency of the literature cited implementing the paragraph on the role of ERK in regulating PD-1/PD-L1 in the revised version (page 12, line 489).
11) 435-- respstive ligands à respective
We corrected the typo.
12) 436 -- have been isentified à identified
We corrected the typo.
13) 534—This paragraph (ref 81 and 83) is hard to follow and understand each sentence
We rephrased the paragraph to make it easier to follow (page 15, line 590).
14) 534- induce the translocation of calneticulin, a Ca-sensor residing within ER on à onto the cell surface of tumor cells?
We modified the sentence (see reply to point 13) .
15) 540- Here, calreticulin acts as ……, triggering the tumor cell phagocytosis by DCs and the consequent (?) expansion of CD8+ …
We modified the sentence (see reply to point 13).
16) 542 – In parallel, since the pgp present on the tumor cells plasma membrane …. à this sentence is talking about the tumor cells, correct?
We modified the sentence (see reply to point 13).
17) 552 – Using self-assembled …. , that have a high intratumor delivery if the drug by preventing the bone uptake
We corrected the typo.
18) 556 – As it occurs for conventional chemotherapy like doxorubicin, also the anti-tumor efficacy of TKI may be in part mediated by the immune system. –
How does this sentence make sense?
We rephrased the sentence (page 16, line 620).
19) 568 – The combined inhibition of MEK/ERK and Smad3/mTOR à MEK/ERK, Smad3 and mTORC (if its triple combination)
We corrected the sentence (page 16, line 631).
20) 575 --The result is à This results in ….a low activation of NK cells this results in lowering the NK cell activation status, and consequently the enrichment of ….
within the tumor..(?)
We corrected the sentence (page 16, line 638).
21) 579 -- ,further combination therapies based on chemotherapy treatment, targeted-ERK inhibition and immune-therapy are noteworthy to be tested in preclinical models of this resistant tumor type. à Dose this sentence mean like “ ,further strategies with possible combinations of ERK inhibitors and immune-therapy with chemotherapy drugs are noteworthy to be tested…”
We confirmed that the meaning of the sentence. We corrected it in the revised version (page 16, line 641).
21) 7. Conclusions and future perspectives
This section lacks references and contains ambiguous sentences. For example, 654 --In aggressive tumors, the disruption of these vicious circuitries with ERK inhibitors, metabolic modifiers down-regulating ERK activity or inhibitors of extracellular factors controlling ERK, may have a potential utility in combination with conventional chemotherapy, other targeted therapies or immune-therapy.
-- What are these inhibitors?
The section “Conclusions and future perspectives” contains the authors’ point of view, based on the literature reported previously. Hence, we include very few references.
We rewrote the section to make the sentences unambiguous.
22) 615 – inhibiting RTK by specific TKIs
We corrected the sentence.
23) 617 – downstream oncogenic transducers
We corrected the typo.
24) 620 – RTK/BRAF.MEK
We corrected the typo.
25) 622 – different RTKs
We corrected the typo.
26) 624 – or the upstream TKI à RTK (what is it meant? Ligands?)
We apologized for the mistake, we corrected TKI into RTK.
27) 632—KRAS/BRAF/MEK/ERK -- > RAS/BRAF/MEK/ERK
We corrected the typo.
28) Overall, the manuscript tends to omit ref in the major journals, not only as listed above, here’s another relevant paper recently published but not picked up by the authors,
https://science.sciencemag.org/content/362/6421/1416
We included more papers from major journals throughout the revised version, included the paper cited above (page 11, line 452).
29) The section 4 and 5 are pretty long and include diverse points. It would help to have subtitles according to the contents for the each paragraph.
e.g., 435—“ERK activity in (tumor-infiltrating) immune cells”
We divided section 4 and 5 in subsections, as suggested.
30) The white labeling of the molecules in the figures are not pretty visible with weak color background (pink, beige or light green).
We modified the Figures as suggested.
Reviewer 3 Report
This review summarizes the current knowledge of the role of the ERK pathway in resistance to chemotherapy and immune escape.
The authors have done a very comprehensive review of the literature on this topic. The manuscript details study by study all the reported protective effects that have been ascribed to ERK in response to chemotherapy and in the context of tumour immune response.
One main problem is that the authors often just list a series of experiments to prove a point, rather than making a synthesis of what the studies concluded. This is particularly acute in sections 2 and 3. These section should be re-written. Sections 4, 5and 6 are much better written.
The second main problem is that there are too many spelling and syntax errors, and that many sentences are difficult to understand because of the lack of proper syntax or incorrect choice of words (see several examples below). I have made corrections from the beginning of the manuscript but the entire manuscript should be thoroughly checked. The manuscript should undergo extensive English editing.
So overall, the content of the review seems exhaustive, but the form needs considerable editing.
Specific Corrections:
There are a number of minor formatting issues, such as
blank spaces (ex, lanes 38)
Typos:
line 53: cromatin : change to chromatin
lines 122, 133, 144 (twice), 155 (savage?), 259. 286 (twice) 306, 361, 435, 436,
spelling or syntax errors:
line 25, socalled –change to so-called
line 39: “cell cycle checkpoint p21 and p27”, should be changed to “cell cycle checkpoint regulators p21 and p27”,
line 66: overlapped, change to overlapping
line 67: faith should be changed to fate.
missing references:
line 38 – after enumeration of all the factors, a reference is needed.
Line 204 – sentence is missing a reference
Line 295 sentence is missing a reference
Line 535: reference is missing.
Other types of correction:
Line 42, remove “proliferation” , since the sentence starts by “Beyond proliferation”
Sentence lines 117-120 needs re-writing
Sentence lines 129-131 needs re-writing
Line 136 – “on the other hand” cannot be used in that sentence
Line 139-142 – this sentence is difficult to understand- needs re-writing.
Lines 155-160: sentence and conclusions don’t make sense and are don’t seem to reflect the conclusions of the paper cited in reference.
Line 168: “In accordance with the profile of pleiotropic kinases” – does not make sense
Line 206: what are “sensitive” prostate tumours?
Lines 419-422: very long sentence difficult to understand. Needs to be broken up in shorter sentences
Lines 564-565: sentence does not make sense.
Other:
Figures 2 and 3: the graphic should be modified, it is difficult to read the legend in the light pink drawings and in the light green drawings.
Line 356: what are : inhibitory immune checkpoints (ICP)? This is not defined, and comes back later such as in in Section 5 line 434
Line 425: immune-activae?
Author Response
Reply to Reviewer 3
This review summarizes the current knowledge of the role of the ERK pathway in resistance to chemotherapy and immune escape.
The authors have done a very comprehensive review of the literature on this topic. The manuscript details study by study all the reported protective effects that have been ascribed to ERK in response to chemotherapy and in the context of tumour immune response.
One main problem is that the authors often just list a series of experiments to prove a point, rather than making a synthesis of what the studies concluded. This is particularly acute in sections 2 and 3. These section should be re-written. Sections 4, 5and 6 are much better written.
The second main problem is that there are too many spelling and syntax errors, and that many sentences are difficult to understand because of the lack of proper syntax or incorrect choice of words (see several examples below). I have made corrections from the beginning of the manuscript but the entire manuscript should be thoroughly checked. The manuscript should undergo extensive English editing.
So overall, the content of the review seems exhaustive, but the form needs considerable editing.
Since Sections 2 and 3 were a premise to the main topics of the review, we did not present an in depth review of the experimental data reported in these Sections in the previous version. However, following the Reviewer’s suggestion, we expanded and rewrote Sections 2 and 3 in a more critical and consistent way in the new version.
We extensively revised the manuscript, corrected spelling and syntax errors, and edited English style.
Specific Corrections:
There are a number of minor formatting issues, such as
1) blank spaces (ex, lanes 38)
We removed the blank spaces throughout the manuscript.
2) Typos: line 53: cromatin : change to chromatin
We corrected the typo.
3) lines 122, 133, 144 (twice), 155 (savage?), 259. 286 (twice) 306, 361, 435, 436
We corrected the typos.
4) spelling or syntax errors: line 25, socalled –change to so-called
We corrected the word.
5) line 39: “cell cycle checkpoint p21 and p27”, should be changed to “cell cycle checkpoint regulators p21 and p27”
We modified the sentence as required (page 1, line 36).
6) line 66: overlapped, change to overlapping
We corrected the word.
7) line 67: faith should be changed to fate.
We corrected the word.
8) missing references: line 38 – after enumeration of all the factors, a reference is needed.
We added the missing references.
9) Line 204 – sentence is missing a reference
We added the missing reference
10) Line 295 sentence is missing a reference
We added the missing reference.
11) Line 535: reference is missing.
We added the missing references.
12) Other types of correction: Line 42, remove “proliferation” , since the sentence starts by “Beyond proliferation”
We modified the sentence as required.
13) Sentence lines 117-120 needs re-writing
We rewrote the sentence (page 4, line 171).
14) Sentence lines 129-131 needs re-writing
We rewrote the sentence (page 4, line 176).
15) Line 136 – “on the other hand” cannot be used in that sentence
We rephrased the sentence (page 5, line 187).
16) Line 139-142 – this sentence is difficult to understand- needs re-writing.
We rephrased the sentence (page 5, line 192).
17) Lines 155-160: sentence and conclusions don’t make sense and are don’t seem to reflect the conclusions of the paper cited in reference.
We apologize for having reporting the main message of the paper in a not correct way. We rephrased the sentence in the revised manuscript (page 5, line 206).
18) Line 168: “In accordance with the profile of pleiotropic kinases” – does not make sense
We recognize that the meaning of “In accordance with the profile of pleiotropic kinases” is not clear. We deleted it in the revised version.
19) Line 206: what are “sensitive” prostate tumours?
We changed “sensitive” into “chemosensitive”.
20) Lines 419-422: very long sentence difficult to understand. Needs to be broken up in shorter sentences
We rephrased the sentence (page 12, line 481).
21) Lines 564-565: sentence does not make sense.
We rephrased the sentence, explaining that the high activity of ERK in resistant cancers has been reported to increase the expression of ICP-L. This leads to the hypothesis that higher doses of ICP/ICP-L inhibitors are required against chemoresistant tumors with activated ERK (page 16, line 626).
22) Other: Figures 2 and 3: the graphic should be modified, it is difficult to read the legend in the light pink drawings and in the light green drawings.
We modified the figures as suggested.
23) Line 356: what are : inhibitory immune checkpoints (ICP)? This is not defined, and comes back later such as in in Section 5 line 434
We defined the concept of immune-checkpoints when they are first mentioned in the text (page 10, line 415).
24) Line 425: immune-activae?
We corrected the typo.
Round 2
Reviewer 1 Report
Thank you for making these extensive revisions. The paper is vastly improved.
Reviewer 2 Report
The authors made a significant improvement.